# CROSS-EMBODIMENT DEXTEROUS GRASPING WITH REINFORCEMENT LEARNING

**Haoqi Yuan**[1], **Bohan Zhou**[1], **Yuhui Fu**[1], **Zongqing Lu**[1,2†]

[1]School of Computer Science, Peking University
[2]Beijing Academy of Artificial Intelligence

## ABSTRACT

Dexterous hands exhibit significant potential for complex real-world grasping tasks. While recent studies have primarily focused on learning policies for specific robotic hands, the development of a universal policy that controls diverse dexterous hands remains largely unexplored. In this work, we study the learning of cross-embodiment dexterous grasping policies using reinforcement learning (RL). Inspired by the capability of human hands to control various dexterous hands through teleoperation, we propose a universal action space based on the human hand's eigengrasps. The policy outputs eigengrasp actions that are then converted into specific joint actions for each robot hand through a retargeting mapping. We simplify the robot hand's proprioception to include only the positions of fingertips and the palm, offering a unified observation space across different robot hands. Our approach demonstrates an 80% success rate in grasping objects from the YCB dataset across four distinct embodiments using a single vision-based policy. Additionally, our policy exhibits zero-shot generalization to two previously unseen embodiments and significant improvement in efficient finetuning. For further details and videos, visit our project page.

## 1 INTRODUCTION

Robotic dexterous grasping (Bicchi, 2000; Duan et al., 2021) has been studied for decades, establishing a foundation for embodied agents to interact with the world through robotic hands. Recent advances have focused on grasping various objects using multi-fingered dexterous hands, employing data-driven methods (Qin et al., 2022b; Mandikal & Grauman, 2022; Liu et al., 2023) or reinforcement learning (RL) (Xu et al., 2023; Wan et al., 2023; Wu et al., 2024b), with real-world deployment of the learned policies (Qin et al., 2023a; Agarwal et al., 2023). However, existing approaches typically learn policies tailored to specific dexterous hands, such as ShadowHand. Developing a policy for a new embodiment often necessitates substantial data collection or costly simulations and hyperparameter tuning in RL.

In this paper, we aim to develop a cross-embodiment dexterous grasping policy (**CrossDex**) that is applicable to various dexterous hands. Cross-embodiment learning leverages shared structural features among different robots to acquire generalized skills, thus facilitating enhanced generalization to new robotic embodiments. Recent research has utilized these shared structures to learn cross-embodiment policies for different robot arms (Chen et al., 2024), robots with varied morphologies (Liu et al., 2022; Hejna III et al., 2021), and embodiments for navigation and manipulation (Yang et al., 2024; Doshi et al., 2024). However, applying cross-embodiment learning to diverse dexterous hands remains largely unexplored and presents several challenges. Dexterous hands feature high degrees-of-freedom (DoFs) and vary in the number of fingers and DoFs, which complicates the unification of the action space for control across different robots. For instance, it is not straightforward to map a control command for a 22-DoF, 5-fingered ShadowHand to a 16-DoF, 4-fingered LEAP Hand (Shaw et al., 2023). Moreover, even if actions could be aligned among different hands, their varied sizes and shapes make it challenging to implement a single policy in contact-rich grasping tasks. For example, Patel & Song (2024) develops a policy for LEAP Hands with varied link

---

[†]Correspondence to Zongqing Lu <zongqing.lu@pku.edu.cn>.

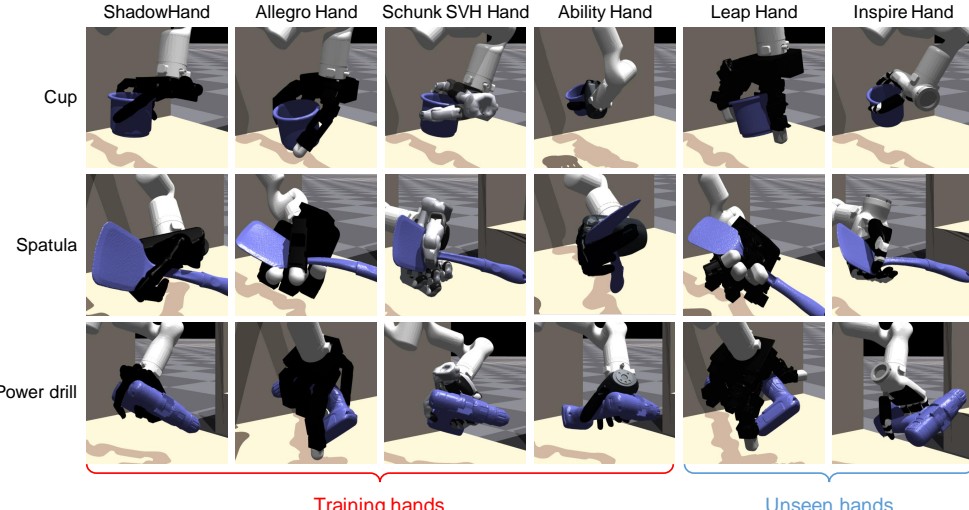

**Figure 1:** We propose CrossDex, learning a cross-embodiment policy for dexterous grasping. The learned RL policy can grasp diverse objects with a variety of dexterous hands and transfer to hands not seen during training.

morphology using graph neural networks (Scarselli et al., 2008), but this approach struggles to generalize between LEAP Hand and Allegro Hand, which, despite their structural similarities, differ in the joint limits of DoFs and shapes of links.

We take inspiration from how humans teleoperate robot hands to tackle this challenge. Humans are not only adept at manipulating everyday objects with their own hands but are also capable of transferring such manipulation skills to various robotic dexterous hands. In teleoperation systems (Li et al., 2019; Handa et al., 2020; Qin et al., 2023b), by observing the robot hand and simultaneously adjusting their own hand poses, human operators can remotely control any robotic hand to grasp objects without needing prior knowledge of the specific robot hand or extensive practice. We adopt a human-like policy that can execute actions within the space of human hand poses to "teleoperate" different dexterous hands. In our approach, we use eigengrasps (Ciocarlie et al., 2007) of the MANO (Romero et al., 2022) hand model as the unified action space, which efficiently compresses 45-dimensional hand poses into low-dimensional eigenvectors. These output hand poses are then converted into each dexterous hand's joint positions through a retargeting algorithm (Handa et al., 2020), providing commands for the controller. We propose training neural networks to replace the optimization-based retargeting process, significantly improving the training speed. To unify the hand's proprioception within the observation space, we eliminate the hand-specific joint positions and adopt the 3D positions of fingertips and palms. These positions are crucial in grasping tasks (Handa et al., 2020) and are consistent across different embodiments.

We train the policy to grasp objects from the YCB dataset (Calli et al., 2015) using reinforcement learning within the IsaacGym (Makoviychuk et al., 2021) simulation environment. Following the teacher-student framework (Jia et al., 2022; Wan et al., 2023), we first train individual state-based policies for each object using PPO (Schulman et al., 2017), and then distill these into a vision-based policy using DAgger (Ross et al., 2011) to grasp all objects. To train each policy, we use four different dexterous hands in parallel environments, each attached to a 6-DoF robot arm with its base fixed on a table. Experimental results demonstrate that our CrossDex policy outperforms baseline methods on the four training hands and two hands not seen during training. CrossDex achieves high success rates in grasping YCB objects and exhibits zero-shot generalization to unseen embodiments with a single policy. Additionally, the policy shows a significant increase in learning efficiency on unseen embodiments and objects via finetuning.

Our contributions are summarized as follows:

- We propose cross-embodiment training for robotic dexterous grasping, marking a step towards a universal grasping policy.

- We adopt techniques from teleoperation to construct a universal action space and observation space for policy learning, enabling transfer across various dexterous hands.

- Our experimental results demonstrate the superior training performance and generalization capabilities of CrossDex.

## 2 RELATED WORK

**Dexterous grasping.** Grasping is a fundamental skill that enables robotic manipulators to interact with the real-world environment. Multi-fingered dexterous hands (Pons et al., 1999; Kappassov et al., 2015; Shaw et al., 2023), in contrast to traditional parallel grippers (Yi et al., 2002; Hu et al., 2019; Fang et al., 2020), offer advanced hardware capabilities. However, the complexity of using dexterous hands to grasp objects brings significant challenges. To address how to grasp, a substantial number of studies focus on generating grasping poses (Zhu et al., 2021; Shao et al., 2020; Li et al., 2023; Liu et al., 2023) and constructing grasping datasets (Wang et al., 2023; Chao et al., 2021). Real-world applications require not only static grasping poses but also closed-loop grasping policies that can execute complete trajectories. Recent studies utilize learning from demonstrations (Qin et al., 2022b; Mandikal & Grauman, 2022; Qin et al., 2022a; Zhou et al., 2024) and deep reinforcement learning (RL) (Xu et al., 2023; Wan et al., 2023; Wu et al., 2024b; Qin et al., 2023a; Kannan et al., 2023) to develop dexterous grasping policies. RL offers distinct advantages due to its independence from human supervision and its scalability to numerous objects (Xu et al., 2023; Huang et al., 2024; Zhang et al., 2025). In this paper, we study cross-embodiment dexterous grasping using RL, marking a significant step towards a universal policy.

**Cross-Embodiment Learning** lays the foundation for developing generalized agents by utilizing data across various embodiments. In the field of embodiment transfer (Niu et al., 2024), leveraging data from source embodiments to learn target embodiment policies facilitates data-efficient transfer learning. Techniques include modeling mappings between embodiments (Zhang et al., 2021; Chen et al., 2024), learning invariant representations (Zakka et al., 2022; Salhotra et al., 2023; Wang et al., 2024), and employing hierarchical learning to share high-level policies (Hejna et al., 2020).

Another line of research focuses on multi-robot training to develop universal, generalizable policies. Studies by Hejna III et al. (2021); Liu et al. (2022); Patel & Song (2024) explore generating varied morphologies and utilizing graph neural networks (Scarselli et al., 2008) to learn embodiment representations. Yu et al. (2023); Bousmalis et al. (2023) use Transformers (Vaswani et al., 2017) to adapt policies to different embodiments in-context. Recent efforts (Doshi et al., 2024; Yang et al., 2024; Xu et al., 2024) aim to train a universal policy that accommodates embodiments with diverse functionalities, such as combining manipulation and navigation tasks.

However, existing research has not yet developed a universal policy for various dexterous hands. GET-Zero (Patel & Song, 2024) learns a policy for in-hand reorientation tasks using different morphological variants of LEAP Hand (Shaw et al., 2023), but this method cannot generalize across different types of dexterous hands. Our research is the first to introduce a universal grasping policy for diverse dexterous hands, achieving zero-shot generalization on hands not seen during training.

**Hand Motion Retargeting** is primarily studied within the context of robotic teleoperation (Antotsiou et al., 2018; Qin et al., 2023b; Fu et al., 2024), where detected human hand poses are transformed into target joint positions of robotic hands. Practical approaches include direct mapping (Fu et al., 2024), supervised learning (Li et al., 2019), and optimization for energy functions (Antotsiou et al., 2018; Handa et al., 2020; Sivakumar et al., 2022). DexPilot (Handa et al., 2020), a notable optimization-based method for dexterous hand teleoperation, explicitly considers the relationships between fingertip positions. In our research, we integrate DexPilot with an RL policy, enabling the control of various dexterous hands with a single policy.

## 3 PRELIMINARIES

### 3.1 REINFORCEMENT LEARNING FOR UNIVERSAL DEXTEROUS GRASPING

We consider a tabletop grasping scenario where a dexterous hand is mounted on a 6-DoF robot arm with its base fixed (see Figure 1). The goal is to grasp and lift an object initially placed on the table.

We assume that tasks differ in the dexterous hand $h \in \mathcal{H}$ and the object $\omega \in \Omega$, where $\mathcal{H}$ denotes the embodiment space and $\Omega$ denotes the object set. Each task, characterized by a unique combination of an embodiment and an object, is formulated as a Partially Observable Markov Decision Process (POMDP) $M_{h,\omega} = \langle \mathcal{O}, \mathcal{S}, \mathcal{A}, \mathcal{T}, R, \mathcal{U} \rangle$, which represents the observation space $\mathcal{O}$, the state space $\mathcal{S}$, the action space $\mathcal{A}$, the transition dynamics $\mathcal{T}(s_{t+1}|s_t, a_t)$, the reward function $R(s_t, a_t)$, and the observation emission function $\mathcal{U}(o_t|s_t)$. At each timestep $t$, the agent observes $o_t \in \mathcal{O}$ and takes an action $a_t \in \mathcal{A}$, then receives a reward $r_t = R(s_t, a_t)$. The environment then transitions to the next state $s_{t+1} \sim \mathcal{T}(s_{t+1}|s_t, a_t)$. The objective of **cross-embodiment dexterous grasping** is to maximize the expected return across all tasks $\sum_{h \in \mathcal{H}, \omega \in \Omega} \mathbb{E}\left[\sum_{t=0}^{T-1} \gamma^t r_t\right]$, where $T$ is the time limit and $\gamma$ is a discount factor.

Existing literature (Qin et al., 2023a; Xu et al., 2023; Wu et al., 2024b) on dexterous grasping focuses on using a single dexterous hand ($|\mathcal{H}| = 1$) with $m$ DoFs and $n$ fingers to grasp any object. The observation $o \in \mathcal{O}$ consists of: (1) Robot proprioception, including the joint positions of the arm $\boldsymbol{J}^a \in \mathbb{R}^6$, the joint positions of the hand $\boldsymbol{J}^h \in \mathbb{R}^m$, and the positions of the fingertips and palm $\boldsymbol{x}^h \in \mathbb{R}^{(n+1) \times 3}$; (2) Object perception, which includes the object pose $\boldsymbol{b} \in \mathbb{R}^7$ consisting of the object position and quaternion in simulation and a visual code $\boldsymbol{v}^\omega$ representing the static shape of the object. In the real world, however, we access object information through visual observations. We choose to use the object point cloud $\boldsymbol{p} \in \mathbb{R}^{N \times 3}$, which contains $N$ points captured by cameras. The action $a \in \mathcal{A}$ includes the target joint positions for the arm $\hat{\boldsymbol{J}}^a$ and the target joint positions for the hand $\hat{\boldsymbol{J}}^h$, which are passed to a PD controller for joint torque control. The objective is to learn a **vision-based policy** $\pi_\phi^V\left(a_t|\boldsymbol{J}_t^a, \boldsymbol{J}_t^h, \boldsymbol{x}_t^h, \boldsymbol{p}_t, a_{t-1}\right)$ parameterized by $\phi$ to maximize the expected return across all objects.

Training a vision-based policy for all objects directly using RL encounters optimization challenges inherent in multi-task RL (Yu et al., 2020), as well as the high-dimensionality of point cloud observations, which can slow down the learning process. Recent works (Jia et al., 2022; Xu et al., 2023; Wan et al., 2023) propose a teacher-student framework with curriculum learning to address these issues. We adopt a simplified approach from these studies. First, the object set is clustered into several groups, and a **state-based policy** $\pi_{\psi_i}^S\left(a_t|\boldsymbol{J}_t^a, \boldsymbol{J}_t^h, \boldsymbol{x}_t^h, \boldsymbol{b}_t, \boldsymbol{v}^\omega, a_{t-1}\right)$ parameterized by $\psi_i$ is trained using RL for each group $i$. Then, all state-based policies $\{\pi_{\psi_i}^S\}$ are distilled into a single vision-based policy $\pi_\phi^V$ using DAgger (Ross et al., 2011), an online imitation learning algorithm. In our research, we leverage this framework to address the challenge of learning a universal vision-based dexterous grasping policy for various embodiments and objects.

## 3.2 HAND POSE MODELING AND RETARGETING

The main challenge in cross-embodiment learning arises from the misalignment of hand joint positions $\boldsymbol{J}^h$ that occurs in both observations and actions among different dexterous hands. This divergence results from differences in dimensions, joint limits, and the functionality of each joint across various embodiments. Then, can we find a shared representation of hand joint positions across these embodiments and establish mappings between them?

The literature on hand teleoperation (Handa et al., 2020; Qin et al., 2023b) suggests that human hand poses can effectively represent poses for dexterous hands. A human hand pose is represented by $(\theta, \beta)$, where $\theta \in \mathbb{R}^{48}$ denotes the 16 axis angles of the hand joints and wrist, while $\beta \in \mathbb{R}^{10}$ denotes the hand shape. Using the MANO hand model (Romero et al., 2022), we can derive the 3D positions of 21 keypoints on the hand from the hand pose: $\boldsymbol{x}^M = \text{MANO}(\theta, \beta), \boldsymbol{x}^M \in \mathbb{R}^{21 \times 3}$. The human hand pose can then be mapped to the joint positions of any dexterous hand through a retargeting process. We adopt optimization-based retargeting methods that optimize $\boldsymbol{J}_t^h$ at each timestep $t$:

$$\min_{\boldsymbol{J}_t^h} S\left(f^h(\boldsymbol{J}_t^h), \boldsymbol{x}_t^M\right) + \|\boldsymbol{J}_t^h - \boldsymbol{J}_{t-1}^h\|^2 \tag{1}$$

$$\text{s.t.} \quad \boldsymbol{J}_{lower}^h \leq \boldsymbol{J}_t^h \leq \boldsymbol{J}_{upper}^h, \tag{2}$$

where $f^h$ is the forward kinematics function of the dexterous hand, providing the positions of the fingertips and palm. $\boldsymbol{J}_{lower}^h$ and $\boldsymbol{J}_{upper}^h$ represent the joint limits of the dexterous hand, and $S$ is a predefined function that measures the similarity between the robot hand pose and the human hand

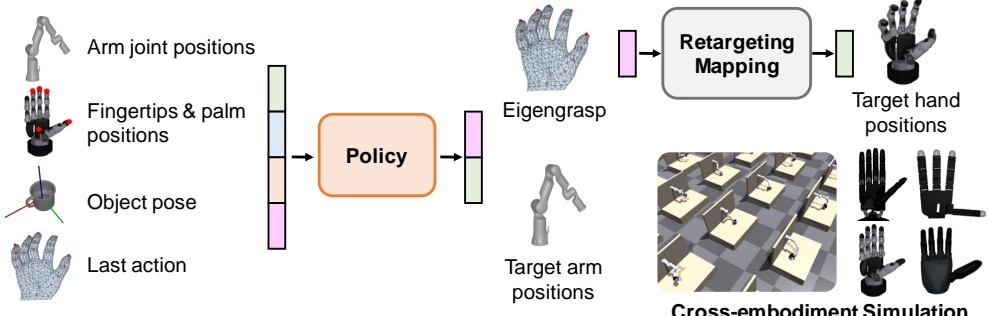

**Figure 2:** CrossDex employs a unified observation and action space to facilitate the learning of a universal policy across various dexterous hands. Rather than relying on joint angles specific to each hand, our policy utilizes the positions of the fingertips and palm to discern the spatial relationship between the hand and the object. Actions are represented using eigengrasps from the MANO hand model, which are mapped to position targets of each hand's PD controller through a retargeting process. This design, akin to teleoperation, enables consistent control across different dexterous hands. The policy is trained using reinforcement learning within a cross-embodiment simulation environment built on IsaacGym. To learn a vision-based policy, we substitute the object pose in this pipeline with the object's point cloud.

pose. For example, DexPilot (Handa et al., 2020) evaluates the distances of the relative positions for each pair of fingertips of the robot hand and the human hand within $S$. This problem can be solved using quadratic programming algorithms. Thus, we can leverage the human hand pose $(\theta, \beta)$ as a universal representation for dexterous hand poses.

# 4 METHOD

In this section, we present the proposed method, CrossDex, which utilizes shared representations of actions and observations to learn a cross-embodiment policy for dexterous grasping. Figure 2 provides an overview of our approach.

## 4.1 EIGENGRASP ACTIONS

We propose using human hand poses as a universal action interface for dexterous hands. The output human hand pose is converted to target joint positions for each dexterous hand through retargeting. Since a human hand pose can be translated into similar grasping poses across various dexterous hands, these actions have the potential to generalize across embodiments, sharing similar trajectories for grasping each object.

However, the dimensionality of human hand poses is significantly higher than the DoFs of most dexterous hands, which can make RL less efficient. Additionally, because the joints of human hands do not move independently and the limits of each joint are unknown, defining the policy's output space becomes challenging. We aim for the policy's output space to be a compact subspace of human hand poses that contains meaningful and natural poses. To achieve this, we adopt eigengrasps (Ciocarlie et al., 2007) to compress human hand poses.

Given a dataset $\mathcal{D} = \{\theta_i\}$ capturing human hand poses in daily life, where each $\theta_i$ represents the axis angles of a hand pose (with $\beta = 0$ to disregard the variability in shapes), we apply PCA to the dataset and retain the first-$k$ eigenvectors, $\{e_i\}_{i=1}^{k}$, known as eigengrasps. Each eigengrasp represents a distinctive hand pose, and we can generate diverse, novel hand poses by linearly combining these eigengrasps. In addition to the arm actions, our cross-embodiment policy outputs $k$-dimensional hand actions $\boldsymbol{w} = (w_1, \cdots, w_k)$, which represent a weighted sum of eigengrasps: $\theta = \sum_{i=1}^{k} w_i \boldsymbol{e}_i$. The hand pose $\theta$ is then converted into keypoint positions using MANO and subsequently mapped to the target joint positions of each dexterous hand via DexPilot retargeting.

[Agarwal et al. (2023)](#) also used eigengrasps to improve RL for dexterous manipulation. While their work applies eigengrasps in the LEAP Hand's action space, we leverage eigengrasps of human hand poses as a universal action interface, enabling cross-embodiment control through retargeting.

## 4.2 A UNIFIED OBSERVATION SPACE

The hand joint positions $\boldsymbol{J}^h$ within the observation space, which require embodiment-specific interpretation, are likely to hinder the policy when adapting to new embodiments with unknown proprioception structures. Therefore, we simplify the observation by discarding the hand joint positions, retaining only the fingertips and wrist positions $\boldsymbol{x}^h$. These positions are also used in the retargeting process, as shown in Equation 1, providing shared hand state representations across different embodiments. Notably, fingertips and palm positions are crucial for grasping tasks, as they enable the agent to effectively infer actions and value based on the relative positions of the object and the hand.

In practice, we use the concatenated 3D positions of the palm and the first four fingers: thumb, index, middle, and ring. For five-fingered hands, we omit the little finger's position. This simplification is practical as the little finger is typically less critical in grasping tasks. For hands less than four fingers, the method can be generalized by setting the unused finger positions to zero in the observations. In principle, this approach extends to any hands with up to N fingers, maintaining compatibility by zero-padding unused positions.

At this point, we establish the structures for cross-embodiment policies. Our state-based policy takes the form $\pi_\psi^S\left(\hat{\boldsymbol{J}}_t^a, \boldsymbol{w}_t | \boldsymbol{J}_t^a, \boldsymbol{x}_t^h, \boldsymbol{b}_t, \boldsymbol{v}^\omega, \hat{\boldsymbol{J}}_{t-1}^a, \boldsymbol{w}_{t-1}\right)$, while the vision-based policy takes the form $\pi_\phi^V\left(\hat{\boldsymbol{J}}_t^a, \boldsymbol{w}_t | \boldsymbol{J}_t^a, \boldsymbol{x}_t^h, \boldsymbol{p}_t, \hat{\boldsymbol{J}}_{t-1}^a, \boldsymbol{w}_{t-1}\right)$, where $\boldsymbol{w}_t \in \mathbb{R}^k$ denotes the eigengrasp action.

## 4.3 MULTI-TASK LEARNING

Following the teacher-student framework, we divide the entire task space of embodiments and objects, $\mathcal{H} \times \Omega$, into groups. We learn state-based policies individually for each group, then distill these policies into a vision-based policy. Our experimental observations, detailed in Section 5.2, suggest that our policy architecture benefits from co-training across embodiments, exhibiting slightly enhanced stability and comparable performance when trained simultaneously on all embodiments compared to individual training on each dexterous hand. Therefore, we choose to divide only the object set for training the state-based policies.

We establish cross-embodiment simulation environments using IsaacGym ([Makoviychuk et al., 2021](#)) to train policies. During the teacher learning phase, we train a state-based policy $\pi_{\psi_\omega}^S$ for each object $\omega$ using PPO ([Schulman et al., 2017](#)) within parallel environments that iterate through all dexterous hands. In the distillation phase, we set up parallel environments for every combination of objects and hands. Here, we iteratively use the vision-based policy $\pi_\phi^V$ to collect trajectories. Actions in these trajectories are then relabeled using the learned teacher policies $\{\pi_{\psi_\omega}^S\}_{\omega \in \Omega}$, and we optimize $\phi$ with behavior cloning.

While the parallel environments in IsaacGym run at a frame rate of tens of thousands of fps, the retargeting process, which involves iterative optimization, runs at only 300 fps, significantly slowing down the training process. Fortunately, due to the continuous nature of the retargeting mappings from the human hand to dexterous hands, we propose to use neural networks to accelerate the retargeting process. For each dexterous hand $h$, we use MANO and the retargeting algorithm to label all samples $\theta_i \in \mathcal{D}$ from the dataset of human hand poses with the corresponding joint positions of the dexterous hand $\boldsymbol{J}_i^h$. Then, we train a neural network $P_\xi^h$ parameterized by $\xi$ to fit the mapping from $\theta_i$ to $\boldsymbol{J}_i^h$. These trained retargeting networks $\{P_\xi^h\}_{h \in \mathcal{H}}$ enable batch computations within parallel environments, enhancing overall training efficiency.

Due to the limited number of dexterous hand embodiments available, we incorporate some randomization to enrich the embodiment space for training, aiming at enhancing the robustness and transferability of the policy. Specifically, we add Gaussian noise to the origin position of the joint connecting the hand and the arm, which encourages the policy to adapt to variations in the hand attachment.

**Table 1:** Comparison of grasping success rates for YCB objects between CrossDex and baseline methods. The 'Training hands' columns show average performance on the four hands provided during training, while the 'Unseen hands' columns show average zero-shot performance on two hands not seen during training. 'State' denotes state-based RL policies, and 'Vision' denotes vision-based policies after multi-task distillation.

| Method | Training hands (State) | Training hands (Vision) | Unseen hands (State) | Unseen hands (Vision) |
|---|---|---|---|---|
| MT-Raw-OA | **0.914** | 0.782 | 0.054 | 0.162 |
| MT-Raw-A | 0.823 | 0.728 | 0.272 | 0.210 |
| MT-Raw-O | 0.884 | 0.779 | 0.046 | 0.145 |
| CrossDex | 0.885 | **0.800** | **0.391** | **0.352** |

We summarize our training process in Appendix A. Further implementation details can be found in Appendix B.

## 5 EXPERIMENTS

### 5.1 TASKS SETUP

We evaluate CrossDex on 45 daily objects from the YCB dataset (Calli et al., 2015) and 6 dexterous hands, with URDFs provided by Ding et al. (2024). We use four of these dexterous hands in training and the remaining two – a 16-DoF, 4-fingered LEAP Hand and a 12-DoF, 5-fingered Inspire Hand – are reserved for testing the model's generalization capabilities. In simulation, we implement a unified reward to train across all objects and dexterous hands:

$$r^{total} = r^{dis} + r^{height} + r^{xy} + r^{success}, \tag{3}$$

where $r^{dis}$ encourages to minimize the distance between the fingertips and the palm to the object, $r^{height}$ encourages to lift the object, $r^{xy}$ discourages horizontal object displacement, and $r^{success}$ indicates successful task completion. Success is defined as maintaining the object at a specific height for 30 steps while keeping the hand close to the object. The episode ends either when the task is completed or the object falls off the table. Detailed descriptions of the tasks and simulation settings are provided in Appendix B.

To obtain eigengrasps and train the retargeting networks, we use the GRAB dataset (Taheri et al., 2020), which includes 1.6M frames depicting human hands interacting with various objects. For RL and DAgger, we deploy 8192 parallel environments that encompass combinations of all the required objects and the four training hands. Additional training details can be found in Appendix B.

### 5.2 CROSS-EMBODIMENT LEARNING

To evaluate the performance of our method, we compare it against multi-task RL methods for cross-embodiment learning. Highlighting our principal technical contribution of unifying observations and actions for dexterous hands, we introduce **MT-Raw-OA**. This baseline trains policies using the complete proprioception of the robot and the original hand actions of target joint positions. To align the dimensions of hand joint positions across different hands, we pad them to a fixed dimension with zeros. To diminish the ambiguity in such observations and actions, we include a one-hot label of the embodiment within observations. Similarly, **MT-Raw-A** uses original actions with our unified observations and **MT-Raw-O** uses original observations but integrates our proposed eigengrasp actions. All baselines are trained using the same teacher-student framework as our method.

Table 1 presents the success rates of these approaches on YCB objects. Our vision-based policy surpasses all baselines on both training hands and unseen hands, demonstrating its superior performance. According to Figure 5 in the Appendix, our vision-based policy achieves success rates greater than 60% on 42 of the 45 objects across the four training hands. This performance highlights its capability to effectively control various embodiments with a single policy. CrossDex particularly excels in zero-shot performance on new hands, showcasing unique transferability of the learned skills across different embodiments.

**Table 2:** Finetuning performance of CrossDex compared to baselines on the unseen LEAP Hand. We evaluate the finetuning of state-based policies and the multi-task vision-based policy for YCB objects. Additionally, we evaluate multi-task finetuning on unseen objects using 55 objects from the GRAB dataset. Training curves for these experiments can be found in Appendix C.2.

| Method | YCB (5 objects, state) | YCB (multi-task, vision) | GRAB (multi-task, vision) |
|---|---|---|---|
| No-Pretrain | 0.758±0.122 | 0.436±0.159 | 0.313±0.373 |
| MT-Raw-OA | 0.701±0.002 | 0.417±0.007 | 0.651±0.007 |
| MT-Raw-A | 0.798±0.002 | 0.390±0.005 | 0.655±0.006 |
| MT-Raw-O | 0.708±0.014 | 0.385±0.003 | 0.616±0.018 |
| CrossDex | **0.872**±0.013 | **0.643**±0.009 | **0.740**±0.009 |

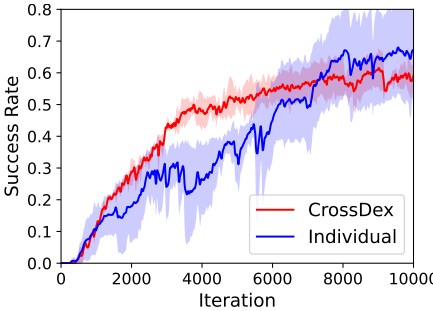

**Figure 3:** Comparison of RL training curves: cross-embodiment learning vs. average performance of individual training.

**Table 3:** Abaltion study examining the selection of optimization objectives used in retargeting. "Position retargeting" focuses on optimizing the absolute positions of the fingertips and palm. "Vector retargeting" focuses on optimizing the relative positions from the palm to each fingertip. "DexPilot" includes optimization of the relative positions between the fingertips.

| Retarget | Training hands | Unseen hands |
|---|---|---|
| Position | 0.884±0.030 | 0.500±0.037 |
| Vector | 0.841±0.042 | 0.435±0.047 |
| DexPilot | 0.892±0.005 | 0.482±0.043 |

MT-Raw-OA and MT-Raw-O demonstrate high performance in state-based training. However, their performance decreases a lot when distilled into a vision-based policy. The primary reason is that these approaches incorporate embodiment labels in observations during state-based training, which makes the policy to rely on this information. During distillation, to enhance generalization across embodiments, we remove these robot labels in the vision-based policy, resulting in less informative observations. Consequently, on unseen hands, these two methods perform worse with state-based policies than with vision-based policies, as the state-based policies utilize the specific unseen label of the new robot, whereas vision-based policies discard this information to enhance generalization.

We notice that even when using the raw action space (MT-Raw-OA and MT-Raw-A), the policy generalizes to unseen hands, albeit with a lower success rate. This generalization is facilitated by the careful alignment of the URDFs for all dexterous hands, ensuring that the arrangement of joint position elements (from the thumb to the little finger) and the corresponding movements (opening the hand with increasing values and closing with decreasing values) are consistent. This manual alignment underperforms our method that leverages human hand eigengrasps to effectively align embodiments.

All methods achieve success rates exceeding 80% during state-based training, demonstrating the effectiveness of jointly training various embodiments. Figure 3 illustrates the training curves of CrossDex alongside those from training each dexterous hand individually. We observe that cross-embodiment training slightly enhances training efficiency and stability compared to individual training, while achieving comparable performance. This observation supports our approach of co-training across embodiments rather than dividing the embodiment space for individual training.

## 5.3 EMBODIMENT TRANSFER

The zero-shot performance on unseen hands, as shown in Table 1, highlights the cross-embodiment tranferability of the learned policy. Such capabilities not only allow for the direct deployment of the policy to new hands but also facilitate efficient finetuning by treating the policy as a pre-trained

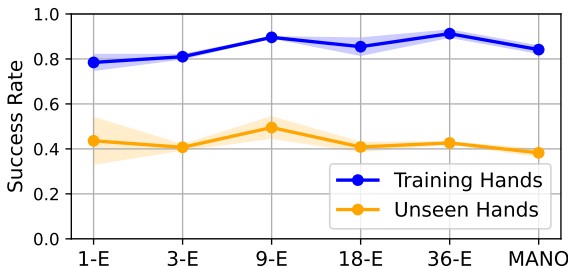

**Figure 4:** Abaltion study on the eigengrasp action space. "MANO" refers to using raw axis angles of the MANO hand model as actions. "$k$-E" refers to using the first-$k$ eigengrasps as actions. Results show average success rates across five YCB objects.

**Table 4:** Ablation study on embodiment randomization and the inclusion of embodiment-specific observations. "Rand." refers to applying randomization to the hand mount position. "Obs." refers to including one-hot robot labels and the randomized positions in the observations.

| Rand. | Obs. | Training hands (State) | Training hands (Vision) | Unseen hands (State) | Unseen hands (Vision) |
|---|---|---|---|---|---|
| ✗ | ✓ | 0.834 | 0.738 | 0.044 | 0.146 |
| ✗ | ✗ | 0.836 | 0.735 | 0.259 | 0.209 |
| ✓ | ✓ | 0.840 | 0.726 | 0.040 | 0.187 |
| ✓ | ✗ | **0.885** | **0.800** | **0.391** | **0.352** |

model for universal grasping skills across all embodiments. We conduct RL finetuning experiments using a simple yet effective approach based on PPO. We initialize the finetuning policy $\pi$ with the pre-trained CrossDex policy $\pi_0$, and during each training iteration, we update $\pi$ using a weighted sum of the surrogate loss from PPO and the KL divergence to the pre-trained policy, $D_{KL}(\pi||\pi_0)$, to mitigate forgetting.

The results are presented in Table 2 and detailed training curves are provided in Appendix C.2. These results include finetuning state-based policies on the unseen hand, finetuning the multi-task vision-based policy on the unseen hand with all objects, and finetuning the vision-based policy to learn 55 unseen objects from the GRAB dataset (Taheri et al., 2020). We observe that CrossDex offers a superior pre-trained policy compared to other methods, excelling in all finetuning tasks involving an unseen hand. Notably, while a policy learned from scratch (No-Pretrain) struggles in multi-task learning, finetuning the CrossDex policy proves to be a promising approach to learn tasks across all objects simultaneously. Furthermore, finetuning approaches demonstrate enhanced stability compared to learning from scratch, as evidenced by the reduced variation and smoother training curves.

## 5.4 ABLATION STUDY

We conduct ablation studies to assess the impact of key components in CrossDex, particularly focusing on the unified actions and observations.

**Retargeting Methods:** In our main results, we use DexPilot retargeting to map eigengrasp actions to robot joint positions. To evaluate it, we compare DexPilot with two other optimization-based retargeting methods by training state-based policies on 5 randomly chosen YCB objects. As shown in Table 3, all retargeting approaches yield similar training and test performance, which underscores the robustness of CrossDex to the choice of retargeting algorithm.

**Eigengrasp Actions:** We explore the importance of using eigengrasps and the impact of varying the number of eigenvectors $k$. Results presented in Figure 4 indicate that modifying $k$ from 1 to 36 leads to consistent training and test performance, highlighting our method's insensitivity to changes in $k$. However, CrossDex-1-E shows decreased training performance, likely due to the limited capacity of

the action space. CrossDex-MANO, which uses the 45-dimensional axis angles of the human hand as actions, performs worse on zero-shot adaptation than the configurations using varied numbers of eigengrasps.

**Embodiment Randomization and Observations:** Table 4 investigates whether applying embodiment randomization or adding embodiment-specific information to observations affects performance. Incorporating randomization appears to improve performance across both training and test hands, suggesting greater generalizability of the policy. However, including embodiment specific observations increases the performance gap between state-based and vision-based policies on training hands, as these observations benefit state-based policies but are unavailable for vision-based policies. Moreover, these observations lead to poorer generalization to unseen hands, supporting our argument for unifying observations for cross-embodiment learning.

## 5.5 REAL-WORLD EXPERIMENTS

We establish a real-world setup using a RealMan RM65 robot arm paired with a LEAP Hand. The object point cloud is captured using Intel RealSense D435i RGBD cameras. We provide video records of the real-world test on our project page.

## 6 CONCLUSION AND LIMITATIONS

In this paper, we propose CrossDex to address the challenges in cross-embodiment learning for dexterous grasping. Our main technical contributions include unifying hand actions with human hand eigengrasps to control various dexterous hands, utilizing pre-trained neural networks for efficient hand pose retargeting, and proposing a shared embodiment-unaware observation space to enhance generalization. Our experimental results demonstrate that CrossDex facilitates robust training compared to individual embodiment training, successfully controls various hand embodiments with a single policy, and effectively transfers to unseen embodiments through zero-shot generalization and finetuning.

Our work has some limitations that can be addressed in future research. First, we train policies on only four dexterous hands due to limited access to dexterous hand models. Future work should include a wider range of dexterous hand embodiments, which is likely to improve embodiment generalization. Additionally, while this work focuses on grasping tasks, there are many real-world tasks for robotic hands – such as in-hand reorientation, dynamic handover, and functional grasping – that remain unexplored in the literature of cross-embodiment learning. Future work could extend CrossDex to these tasks.

ACKNOWLEDGMENTS

This work was supported by NSFC under Grant 62450001 and 62476008. The authors would like to thank the anonymous reviewers for their valuable comments and advice.

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

## A  ALGORITHM

---

**Algorithm 1:** CrossDex training process.

---

**Input:** Dexterous hand models $\mathcal{H}$; Object set $\Omega$; Human hand pose dataset $\mathcal{D}$; Untrained retargeting networks $\{P_\xi^h\}_{h \in \mathcal{H}}$, state-based policies $\{\pi_{\psi_\omega}^S\}_{\omega \in \Omega}$, and the vision-based policy $\pi_\phi^V$.

**Output:** The learned vision-based policy $\pi_\phi^V$.

**Preprocessing:**

$\{\boldsymbol{e}_i\}_{i=1}^k \leftarrow \text{PCA}(\mathcal{D})$;

**for** $h \in \mathcal{H}$ **do**
> Construct $\mathcal{D}' = \{(\theta_i, \boldsymbol{J}_i^h)\}_{\theta_i \in \mathcal{D}}$ using MANO and DexPilot;
> Train $P_\xi^h$ on $\mathcal{D}'$ to minimize MSE loss.

**Teacher learning:**

**for** $\omega \in \Omega$ **do**
> Create IsaacGym environments for object $\omega$ and all hands;
> **while** *not converge* **do**
>> Get state-based observations $o_t^S$;
>> Sample actions $\hat{\boldsymbol{J}}_t^a, \boldsymbol{w}_t \sim \pi_{\psi_\omega}^S(o_t^S)$;
>> Compute target joint positions $\hat{\boldsymbol{J}}_t^h \leftarrow P_\xi^h(\sum_{i=1}^k \boldsymbol{w}_{t,i} \boldsymbol{e}_i)$;
>> Step the environments, get rewards $r_t$;
>> Save $(o_t^S; \hat{\boldsymbol{J}}_t^a, \boldsymbol{w}_t; o_{t+1}^S; r_t)$ into the PPO buffer;
>> Update $\pi_{\psi_\omega}^S$ using PPO.

**Distillation:**

Create IsaacGym environments for all objects and hands;

**while** *not converge* **do**
> Get state-based observations $o_t^S$ and vision-based observations $o_t^V$;
> Sample actions $\hat{\boldsymbol{J}}_t^a, \boldsymbol{w}_t \sim \pi_\phi^V(o_t^V)$;
> Step the environments;
> Get teacher actions $\hat{a}_t \leftarrow \pi_{\psi_\omega}^S(o_t^S)$ for each object $\omega$;
> Save $(o_t^V; \hat{a}_t)$ into the buffer;
> Update $\pi_\phi^V$ to minimize MSE loss using the buffer.

---

## B  IMPLEMENTATION DETAILS

### B.1  TASKS SETUP

For our experiments, we use objects from the YCB dataset (Calli et al., 2015) to evaluate CrossDex. We exclude objects that cannot be modeled as a rigid bodies, objects composed of more than 30 convex hulls after VHACD decomposition (to fit within the IsaacGym simulation), and objects that are either too large or too flat, necessitating functional grasping rewards (Wu et al., 2024a) outside the scope of our study. The resulting dataset comprises 45 objects, representing a diverse array of everyday items with varied geometries, such as cups, scissors, and boxes. The full object list is available in Figure 5. For testing the finetuning performance on unseen objects, we use 55 objects from the GRAB dataset (Taheri et al., 2020). For state-based finetuning and certain ablation studies, we select five random objects from the YCB dataset, including *mustard bottle, mug, spoon, softball,* and *cup_j*.

Our reward function, adapted from UnidexGrasp++ (Wan et al., 2023), is defined as follows:

$$r^{total} = r^{dis} + r^{height} + r^{xy} + r^{success}.$$

Here, $r^{dis}$ penalizes the weighted L2 distance between the object center and the fingertips and palm:

$$r^{dis} = -2 \times \text{average\_fingertips\_object\_distance} - \text{palm\_object\_distance}.$$

The $r^{height}$ term encourages lifting the object to a specific height:

$$r^{height} = \begin{cases} 0, & \text{if average\_fingertips\_object\_distance} \geq 0.12 \text{ and palm\_object\_distance} \geq 0.15 \\ 0.9 - 2|H - 0.6| + (H - 0.6) + \frac{1}{|H-0.6|+1}, & \text{otherwise,} \end{cases}$$

where $H$ denotes the current height of the object, starting from an initial height of 0.3 meters.

$r^{xy}$ aims to minimize horizontal displacement of the object:

$$r^{xy} = -0.3 \times \|\text{object\_xy} - \text{object\_initial\_xy}\|.$$

Finally, $r^{success}$ awards 200 points upon task completion, defined as maintaining $|H - 0.6| \leq 0.05$ and either average\_fingertips\_object\_distance $\leq 0.12$ or palm\_object\_distance $\leq 0.15$ for $n$ steps. $n$ is set to 30 in test and we increase it to 60 in training to enhance policy robustness.

## B.2 EMBODIMENTS

We use six dexterous hand models provided by Ding et al. (2024), allocating four for training and two for test.

The training embodiments consist of:

- ShadowHand: a 5-fingered hand with 22 DoFs.
- Allegro Hand: a 4-fingered hand with 16 DoFs.
- Schunk SVH Hand: a 5-fingered hand with 20 DoFs.
- Ability Hand: a 5-fingered hand with 10 DoFs.

For testing, we use:

- Leap Hand: a 4-fingered hand with 16 DoFs.
- Inspire Hand: a 5-fingered hand with 12 DoFs.

Refer to Figure 11 for a visualization of the six dexterous hands.

Regarding the robot arm, we employ the 6-DoF RealMan RM65 arm. The base of the arm is mounted on the side panel of the table, and the dexterous hand is fixed to the end-effector, which aligns our hardware configuration.

## B.3 TRAINING RETARGETING NETWORKS

The retargeting networks are designed to map 45-dimensional axis angles of fingers to joint positions for each dexterous hand. For simplicity, the 3-dimensional wrist rotation and the shape parameters of the human hand are set to zero. Each retargeting network is a four-layer MLP with hidden layers of 512 dimensions and ReLU activation functions. Each network is trained for 200 epochs, minimizing the MSE loss. Figure 11 visualizes the performance of the learned retargeting networks.

## B.4 TRAINING POLICIES

Our codebase for RL and DAgger is based on UniDexGrasp (Xu et al., 2023). For each state-based policy, we use five-layer MLPs for both the actor and the critic, featuring hidden layers with dimensions $[1024, 1024, 512, 512]$ and using ELU activation functions. For the vision-based policy, we use a simplified PointNet (Qi et al., 2017) backbone incorporating two 1D convolution layers and two MLP layers to process the object point cloud, which has dimensions of 512. Both the actor and the critic share the output of this backbone.

In state-based RL, we train the policies using PPO across 40,000 iterations within parallel simulations of 8,192 environments. Hyperparameters are provided in Table 5. For vision-based distillation,

we use DAgger for 20,000 iterations within parallel simulations of 16,384 environments. Hyperparameters are provided in Table 6. In our finetuning experiments, state-based finetuning is conducted for 10,000 iterations using 8,192 parallel environments, while vision-based multi-task finetuning is conducted for 20000 iterations using 16,384 parallel environments. Our experiments within environments of 8,192 can be completed on a single NVIDIA RTX 4090 GPU. For the larger scale experiments involving 16,384 environments and PointNet backbones, we use a single NVIDIA A800 GPU.

To acquire object point clouds for vision-based policies, we follow the approach in prior work (Wu et al., 2024b). In simulation, object point clouds are constructed from the objects' mesh data. At each timestep, the point clouds are transformed based on the objects' poses. During training, we apply Farthest Point Sampling (FPS) to sample 512 points, which are then fed into a PointNet to extract features. The PointNet is trained jointly with the policy during the distillation process. In the real world, to acquire similar object point clouds, we use RGB-D cameras and instance segmentation to capture the object point cloud at the first timestep. For subsequent timesteps, we use instance tracking and pose estimation to transform the object point cloud, ensuring consistency of the shape across execution steps. Our approach of constructing object point clouds from mesh data in simulation avoids the need for simulated cameras, which would significantly limit the number of parallel environments. Using this approach, IsaacGym can support thousands of parallel environments, allowing us to train vision-based policies efficiently with 16,384 environments on a single NVIDIA A800 GPU in approximately 10 hours. We have also validated that same performance can be achieved using 4,096 environments on an NVIDIA RTX 4090 GPU, which takes approximately 2 days.

**Table 5:** Hyperparameters of PPO.

| Name | Symbol | Value |
|---|---|---|
| Parallel rollout steps per iteration | -- | 8 |
| Training epochs per iteration | -- | 5 |
| Minibatch size | -- | 16384 |
| Optimizer | -- | Adam |
| Learning rate | $\eta$ | 3e-4 |
| Discount factor | $\gamma$ | 0.96 |
| GAE lambda | $\lambda$ | 0.95 |
| Clip range | $\epsilon$ | 0.2 |

**Table 6:** Hyperparameters of DAgger.

| Name | Symbol | Value |
|---|---|---|
| Parallel rollout steps per iteration | -- | 8 |
| Training epochs per iteration | -- | 5 |
| Minibatch size | -- | 4096 |
| Optimizer | -- | Adam |
| Learning rate | $\eta$ | 3e-4 |

## C  ADDITIONAL RESULTS

### C.1  SUCCESS RATES PER OBJECT

Figure 5 shows the success rates achieved by the learned vision-based policy for each YCB object.

### C.2  TRAINING CURVES

Figure 6, 7, and 8 illustrate the reward vs. training iteration curves for the finetuning experiments.

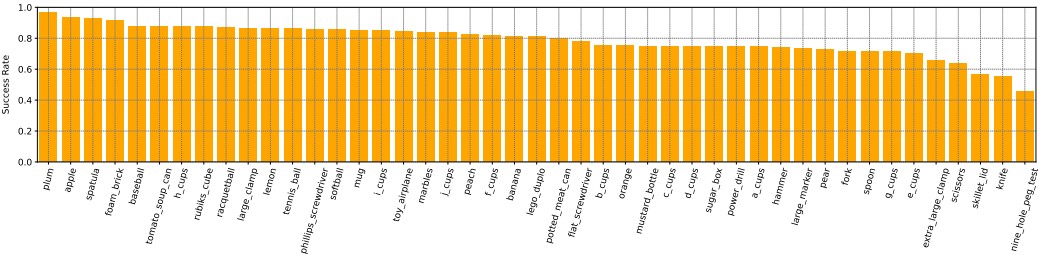

**Figure 5:** Average success rates for each YCB object across various embodiments.

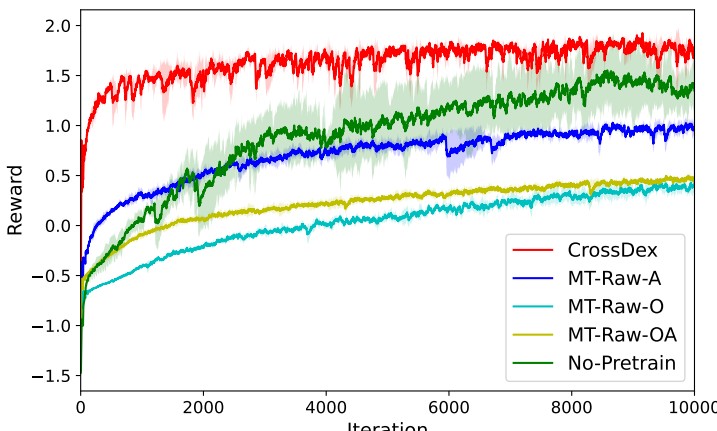

**Figure 6:** Training curves depicting state-based finetuning performance on five YCB objects using the unseen LEAP Hand.

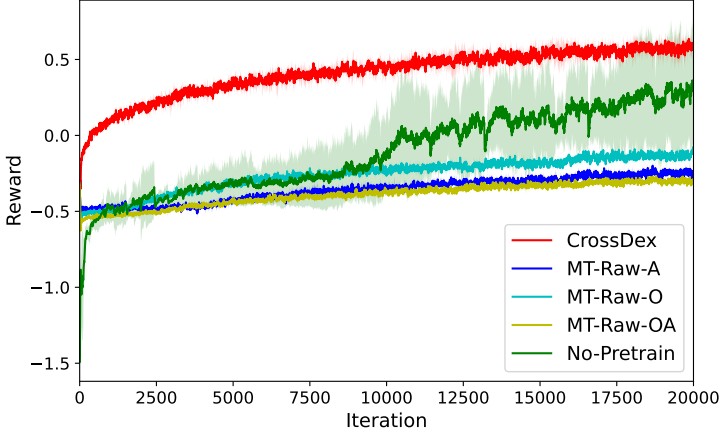

**Figure 7:** Training curves depicting multi-task vision-based finetuning performance on all YCB objects using the unseen LEAP Hand.

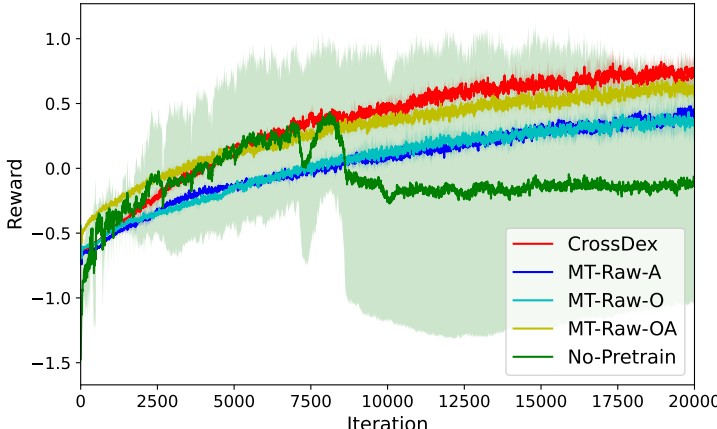

**Figure 8:** Training curves depicting multi-task vision-based finetuning performance on the 55 unseen GRAB objects using the unseen LEAP Hand.

### C.3 TRAINING CURVES FOR DIFFERENT ACTION SPACE CONFIGURATIONS

Figure 9 shows the PPO training curves of CrossDex with different numbers of eigengrasp actions and CrossDex with the 45-dimensional MANO actions.

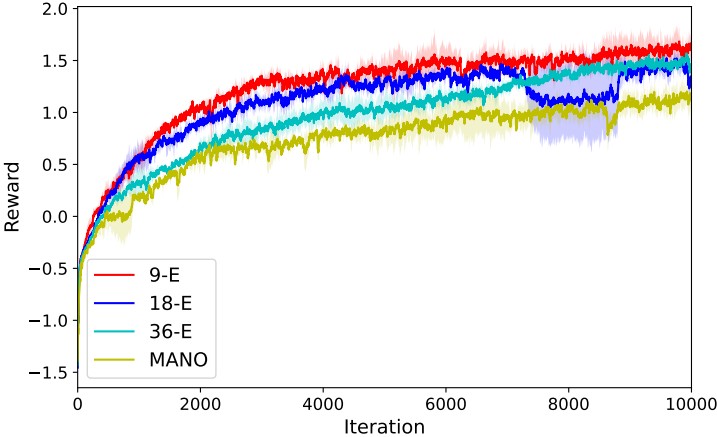

**Figure 9:** Training curves of CrossDex with different human hand action spaces.

## C.4 Visualization of Eigengrasp Actions and Retargeting

Figure 11 displays samples from the eigengrasp action space and the corresponding joint positions of various dexterous hands predicted by our retargeting networks.

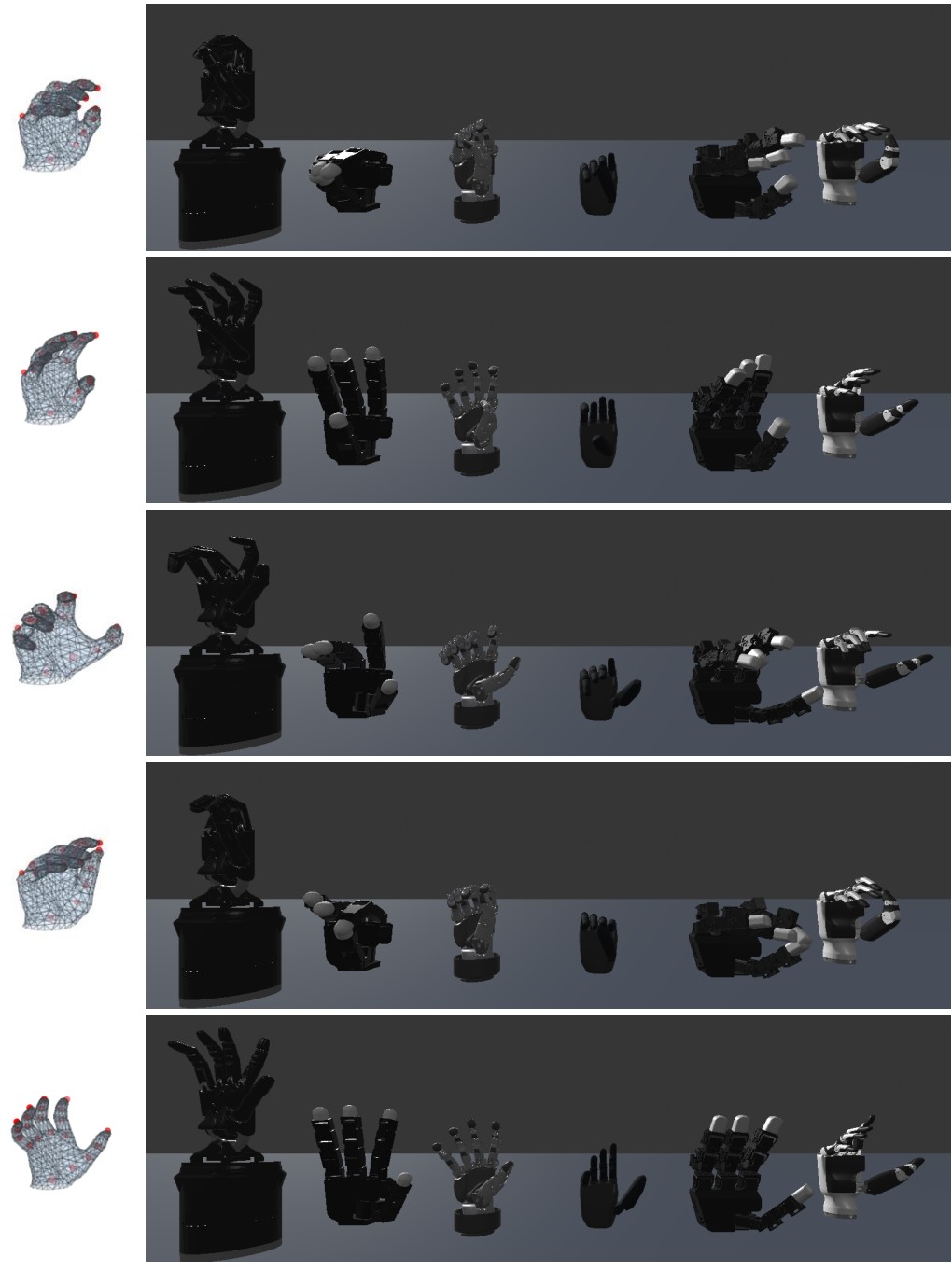

**Figure 10:** Visualization of hand poses for some eigengrasp actions and the corresponding poses of the six dexterous hands as predicted by our retargeting neural networks.

## C.5 THE IMPACT OF THE NUMBER OF TRAINING HANDS

We conduct experiments training CrossDex with different numbers of dexterous hands. The results in Table 7 demonstrate that increasing the number of training hands consistently improves performance on unseen hands, highlighting the potential for further improvement by including more dexterous hands in the training set.

**Table 7:** Performance on the two test hands given different numbers of training hands.

| Number of Training Hands | 1 | 2 | 3 | 4 |
|---|---|---|---|---|
| Success Rate on Unseen Hands | 0.129 | 0.328 | 0.434 | 0.482 |

## C.6 RESULTS FOR A THREE-FINGERED HAND

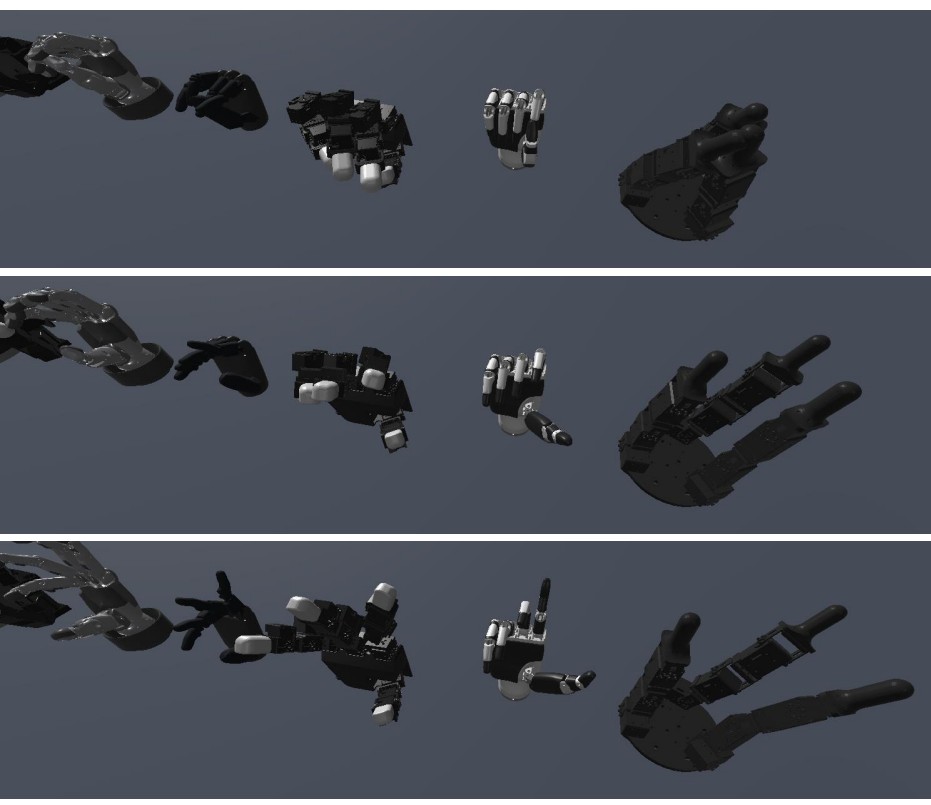

**Figure 11:** Comparison of the retargeted hand poses between the three-fingered D'Claw Gripper and other dexterous hands, with each row corresponding to a specific eigengrasp action.

It is essential to evaluate our method on 3-fingered hands, even though hand retargeting algorithms are less frequently applied to such hands. To explore this, we conduct an experiment using the three-fingered D'Claw Gripper. The retargeting algorithm is adapted by ignoring the ring and small fingers of the human hand and mapping the other three fingers to the robotic fingers. We also use the GRAB dataset to train the retargeting network. We train the CrossDex policy using the four previous training hands along with the D'Claw Gripper on the five randomly selected YCB objects. The results are presented in Table 8.

The results indicate that our method is scalable to the three-fingered D'Claw Gripper, achieving high success rates.

**Table 8:** Performance on the three-fingered D'Claw Gripper trained with the other four hands.

| Hand | ShadowHand | Allegro Hand | Ability Hand | Schunk SVH Hand | D'Claw Gripper |
|---|---|---|---|---|---|
| Num. Fingers | 5 | 4 | 5 | 5 | 3 |
| Success Rate | $0.891 \pm 0.009$ | $0.892 \pm 0.008$ | $0.753 \pm 0.009$ | $0.911 \pm 0.009$ | $0.853 \pm 0.033$ |

In Figure 11, we demonstrate that the retargeting network effectively enables the three-fingered hand to open and close, similar to other dexterous hands. However, we observe lower consistency in achieving specific hand poses compared to anthropomorphic hands, due to the significant differences in morphology. For instance, the D'Claw Gripper struggles with certain grasping poses requiring fingers to curve inward. Future work could address these challenges by developing more tailored retargeting methods for hands with distinct morphologies.

