# OpenReview forum: "Cross-Embodiment Dexterous Grasping with Reinforcement Learning"
_ICLR.cc/2025/Conference — ICLR 2025 Poster_

### Official Review · Reviewer_FoGX · 2024-10-16

**Soundness:** 2
**Presentation:** 2
**Contribution:** 2
**Rating:** 3
**Confidence:** 3

**Summary:**

This paper introduces CrossDex, a method for learning cross-embodiment dexterous grasping by defining a unified, embodiment-independent observation and action space. The action space utilizes eigengrasps of the human hand based on the MANO hand model. To enable efficient training, pre-trained neural networks replace the optimization-based retargeting process from eigengrasps to the joint angles of various embodiments. The policies are trained on grasping tasks using objects from the YCB dataset across different robot hands and evaluated for transferability to unseen embodiments. Additionally, visual policies operating on object point clouds are distilled using DAgger.

**Strengths:**

**Relevance of the Problem**
- The huge diversity of embodiments presents a significant challenge in embodied AI, making research on cross-embodiment learning highly relevant.

**Method**
- Leveraging pre-trained neural networks to approximate the optimization-based retargeting process across different embodiments is a intuitive approach to ensure efficient, parallelized training.

**Presentation**
- The writing is clear, and the logical flow of the paper effectively outlines studied problem and proposed solution, making it intuitive to follow.
- The visual presentation in Figures 1 and 2 is well-executed, providing a clear overview of the simulation setup and the input-output mapping from observations through policy actions to target joint positions.

**Weaknesses:**

**Method and Evaluation**
- The baselines used in the evaluations are ablations of the proposed method and do not appear to be particularly strong for this task.
- Training on 4 embodiments and transferring to 2 novel ones is indeed quite limited, as acknowledged in the limitations section. However, attributing this limitation to 'restricted access to dexterous hand models' is not a compelling justification. Incorporating a simulation with a broader range of dexterous hand models and exploring how the performance of cross-embodiment policies scales with an increasing number of embodiments would add significantly value to the study.
- Details on how the vision policies are trained including camera resolution and a definition of the object point cloud are missing.


**Results and Claims**
- In section 5.3 it is stated that the cross-embodiment transferabiliy of the learned policy allows for direct deployment of the policy to new hands. This statement is not supported by the performance drop of over 50% between seen and unseen embodiments oberved for CrossDex in Table 1.
- In lines 254-255 you state that the "the dimensionality of human hand poses is significantly higher than the DoFs of most
dexterous hands, which can make RL less efficient." However, this high-dimensionality does not seem to be a challenge according to the results in Table 1, where policies using the joint-based action space outperform or perform simlarly to CrossDex in the PPO training stage. Further, related work has shown good performance with PPO in parallized simulators even with very high-dimensional action spaces (https://arxiv.org/abs/2305.12127).
- The comparison to baseline methods does not convincingly support the claim that using eigengrasps is an effective approach to solving this problem. Baseline policies based on joint-level control are not expected to transfer well across embodiments with different and unknown joint limits, especially when there is no information about the function of each joint. Using a baseline that directly controls keypoints in the MANO hand model, rather than using eigengrasps, would be a more appropriate baseline to validate this claim. This approach would more naturally align with the task of positioning the fingers into stable grasping configurations on an object’s surface. Moreover, Figure 9 in the Appendix highlights substantial differences in the resulting joint configurations for the same eigengrasp across different embodiments. For instance, in the first row of the figure, the second hand shows three fingers touching at the tips, whereas in the fifth hand, there is a significant gap between the fingers. These very different grasp configurations raise questions about the reliability of grasp transfer under such variation.
- In the ablation on 'Eigengrasp Actions,' Figure 4 examines the effect of varying the number of eigengrasps on performance it is inferred that the method is relatively insensitive to this change. However, the finding that comparable performance can be achieved with just a single eigengrasp raises important questions. This outcome may reflect a characteristic of the problem being studied, which might not require the nuanced control offered by a multi-fingered hand, rather than demonstrating a strength of the proposed method itself.


**Minor Points on Writing**
- In line 483, "affect" should be "affects".


***

Overall, the fact that baselines not specifically designed for cross-embodiment transfer still achieve non-negligible success raises doubts about the suitability of the proposed approach. Moreover, a success rate below 40% on unseen embodiments does not justify the claim of direct transferability. This calls into question the overall feasibility of the framing. Additionally, the observation that a single eigengrasp performs comparably well on the task suggests that the task itself may not be sufficiently challenging. More complex manipulation problems are needed to meaningfully assess the impact of the proposed action space.

**Questions:**

- What exactly are the observations used by the visual policies, and how are they transferred to the real robot system? In lines 809-810 of the appendix, you mention that the vision policies are trained using DAgger in 16,384 parallel environments, which is double the number of environments used for state-based PPO training. Generating visual observations for such a large number of environments seems particularly challenging, especially given that prior works using student-teacher learning to distill vision policies in Isaac Gym have used two orders of magnitude fewer environments (https://arxiv.org/abs/2211.11744, https://arxiv.org/abs/2111.03043). You also state that "the object point cloud p ∈ R^(N×3), which contains N points captured by cameras." Is this captured by one camera per environment? If so, at what resolution, and how does the object point cloud differ from the point cloud obtained directly from the camera?
- Has a quantitative evaluation been conducted on real-robot deployment?
- In lines 346-347 it is stated that "The episode ends either when the task is completed or the object falls off the table". Are episodes also terminated after a certain number of time-steps?
- Is the object shape or ID represented to the state-based policies? If not, is the grouping of objects intended to reduce variability between them enough to allow learning a single grasping policy? If so, how is this grouping performed, and what are the characteristics of the resulting groups? A clearer explanation of the grouping process and its effectiveness in handling object diversity would help clarify how the policy generalizes across different objects.
- In Figure 3, the training performance of cross-embodiment learning is compared to individual training with the Shadow Hand. Is there a reason to specificallly use the Shadow Hand here, rather than the mean of the performance of all single-embodiment policies?

---

> ### Author Response · Authors · 2024-11-24
> **Thanks for your review! Here, we respond to your comments and address the issues. We hope to hear back from you if you have further questions!**
>
> **Q1.** Baselines are not strong. Using a baseline that directly controls keypoints is necessary.
>
> **A1.** Thank you for your suggestion. Cross-embodiment policy learning for dexterous hands is a less-explored problem, which is why we initially introduced RL baselines with straightforward implementations for comparison. We appreciate that it is necessary to include a baseline controlling the shared actions of fingertip positions. To address this, we have implemented this baseline, named CrossDex-Pos, using the shared observation space of CrossDex and a 15-dimensional action space representing the five fingertip positions relative to the palm position. These actions are mapped to the joint positions of dexterous hands through optimization-based retargeting. Below is the performance comparison of CrossDex-Pos with CrossDex on five randomly sampled YCB objects:
>
> | Method | Training Hands | Unseen Hands |
> |----------------|--------------------|--------------------|
> | CrossDex-Pos |  0.821+-0.034 | 0.423+-0.056 |
> | CrossDex | 0.892+-0.005 | 0.482+-0.043 |
>
> As shown, CrossDex-Pos underperforms compared to CrossDex on both training and unseen hands. This result highlights that while fingertip position control offers a shared action space, it is less effective in achieving high performance and generalization compared to our eigengrasp-based approach.
> We thank you for this valuable suggestion and will incorporate the full results on all YCB objects into the revised paper to strengthen the evaluation of our method.
>
>
> **Q2.** Incorporating a simulation with a broader range of dexterous hand models is significant.
>
> **A2.** We agree that exploring a broader range of dexterous hand models and analyzing how cross-embodiment performance scales with the number of embodiments would provide additional insights. However, we note that the variety of dexterous hand models currently available **in both simulation and the real world is indeed quite limited**.
>
> Some works [1] have attempted to generate diverse embodiments by applying randomizations to existing dexterous hands, such as modifying link lengths or removing links. However, such randomizations are constrained by the link geometries and the overall structural design, which cannot be easily randomized without significantly altering the mechanical feasibility of the hands. Developing realistic, diverse dexterous hand models is a **non-trivial** topic that requires substantial effort in hardware design, simulation fidelity, and control feasibility.
>
> Our work focuses on the core challenge of developing cross-embodiment RL training methods rather than addressing the design of new dexterous hand models. While we recognize the potential value of incorporating more embodiments, we argue that using the limited existing hand models to study cross-embodiment learning is a meaningful and practical first step.
>
> [1] Patel, Austin, and Shuran Song. "GET-Zero: Graph embodiment transformer for zero-shot embodiment generalization." (2024).
>
>
> **Q3.** Questions about the vision policy and the object point cloud.
>
> **A3.** For the point cloud observations, we follow the approach in GraspGF [2]. In simulation, object point clouds are constructed from the objects' mesh data. At each timestep, the point clouds are transformed based on the objects' poses. During training, we apply Farthest Point Sampling (FPS) to sample 512 points, which are then fed into a PointNet to extract features. The PointNet is trained jointly with the policy during the distillation process.
>
> In the real world, to acquire similar object point clouds, we use three RGB-D cameras and instance segmentation to capture the object point cloud at the first timestep. For subsequent timesteps, we use instance tracking and pose estimation to transform the object point cloud, ensuring consistency of the shape across execution steps.
>
> Our approach of constructing object point clouds from mesh data in simulation avoids the need for simulated cameras, which would significantly limit the number of parallel environments. Using this approach, IsaacGym can support thousands of parallel environments, allowing us to train vision-based policies efficiently with 16,384 environments on a single NVIDIA A800 GPU in approximately 10 hours. We have also validated that same performance can be achieved using 4,096 environments on an NVIDIA RTX 4090 GPU, which takes approximately 2 days.
>
> We have updated Appendix B.4 to include these details.
>
> [2] Wu, et al. "Learning score-based grasping primitive for human-assisting dexterous grasping." NeurIPS 2023.
>
> **Q4.** Direct deployment of the policy to new hands should not be claimed due to the performance drop.
>
> **A4.** We acknowledge the generalization gap in our method, which is expected due to the limited number of training embodiments. Improving this will be a focus of future work as more hand models become accessible.

---

> > ### Author Response · Authors · 2024-11-24
> >
> > **Q5.** The high-dimensionality does not seem to be a challenge according to the results in Table 1.
> >
> > **A5.** Our claim refers specifically to the high-dimensionality of **human hand** representations (e.g., $21 \times 3$-dimensional keypoints), which motivates our use of MANO eigengrasps for dimensionality reduction. The baselines in Table 1, such as MT-Raw-OA and MT-Raw-A, use **robot joint position** control with a 22-dimensional action space, rather than the higher-dimensional human hand representations.
> >
> > To address your concern, we have compared the training curves of CrossDex-MANO (45-dimensional actions) with CrossDex using eigengrasps in Appendix C.3. CrossDex-MANO shows worse training efficiency and final performance compared with CrossDex using eigengrasps.
> >
> >
> > **Q6.**  The different grasp configurations in Figure 9 raise questions about the reliability of grasp transfer.
> >
> > **A6.** We respectfully disagree with this point. It is **inherently impossible** to achieve precise alignment of joint configurations across different dexterous hands due to variations in morphology. However, precise alignment is not necessary for successful grasping; similar grasping poses that fulfill the task requirements are sufficient. This principle is well-supported in dexterous teleoperation practices, which inspired our approach.
> >
> > For example, as shown in the first row of Figure 1, despite differences in finger joint configurations across embodiments, the policy generalizes effectively. All hands successfully grasp the cup by using the thumb and index finger to hold its rim, with the remaining fingers providing auxiliary support.
> >
> > **Q7.** About "comparable performance can be achieved with just a single eigengrasp".
> >
> > **A7.** We respectfully clarify that CrossDex with a single eigengrasp (CrossDex-1E) achieves a success rate of 78.45%, but increasing the number of eigengrasps leads to a **significant performance gain** of approximately 10%. This improvement highlights the advantage of utilizing higher-dimensional action spaces to better exploit the dexterity of the hands.
> >
> > To draw an analogy, while a parallel gripper (analogous to CrossDex-1E) can successfully grasp most daily objects, higher-dimensional dexterous hands (analogous to CrossDex-k-E, where k>1) demonstrate enhanced capabilities for handling more challenging or intricate grasping tasks. This gain underscores the strength of our method in leveraging the additional control degrees offered by dexterous hands.
> >
> >
> > **Q8.** Has a quantitative evaluation been conducted on real-robot deployment?
> >
> > **A8.** No, we have not conducted a quantitative evaluation on real-robot deployment. Instead, we provide sim-to-real video demonstrations to show that our simulation and algorithm are capable of transferring to the real world. Quantitative evaluations in the real world are not the primary focus of our study, as real-world performance heavily depends on hardware configurations and object placement conditions. These factors introduce significant variance, which could hinder reproducibility and may not provide meaningful insights for the scope of this work.
> >
> > **Q9.** Are episodes also terminated after a certain number of time-steps?
> >
> > **A9.** Yes, an episode terminates either when the object falls off the table, when the episode exceeds 300 steps, or when the task is successfully completed.
> >
> > **Q10.** How to group objects in state-based training? Is the object shape or ID provided?
> >
> > **A10.** In state-based training, we train one policy for each object individually, eliminating the need to provide object shape information. This approach is feasible given the relatively small size of the YCB dataset. However, we acknowledge that object clustering could be introduced for larger datasets. Existing methods [3,4] for object curriculum could be applied in such cases, though this lies outside the primary focus of our current work.
> >
> > [3] Xu, et al. "Unidexgrasp: Universal robotic dexterous grasping via learning diverse proposal generation and goal-conditioned policy." CVPR, 2023.
> > [4] Wan, et al. "Unidexgrasp++: Improving dexterous grasping policy learning via geometry-aware curriculum and iterative generalist-specialist learning." ICCV, 2023.
> >
> > **Q11.** In Figure 3, why not compare with the mean performance of all hands' policies?
> >
> > **A11.** Thank you for pointing this out. We have updated Figure 3 to include a comparison with the mean performance of individual training across all hands.
> >
> > **Q12.** Minor points.
> >
> > **A12.** Thanks! We have fixed the typo.

---

> > > ### Comment · Reviewer_FoGX · 2024-11-25
> > > **Thank you for your reply**
> > >
> > > I would like to thank the authors for their detailed response to my questions and concerns.
> > > I believe that building a simulation that makes it easy to use a wide variety of dexterous hands is a very valuable contribution to the research community. Extending this effort to more manipulators and possibly adding real robot results to verify the simulator would greatly enhance the contribution of this work.
> > >
> > > Currently, the limited number of robot hands and the performance drop when generalizing to novel hand models make it difficult for me to assess the potential of the presented CrossDex framework. While I believe this is an interesting approach to a highly relevant problem, a larger evaluation leading to strong generalization results is what I am currently missing. I will maintain my score.

---

> ### Author Response · Authors · 2024-11-27
>
> Dear reviewer,
>
> We sincerely thank you for your thoughtful feedback. We acknowledge your primary concern about the limited number of dexterous hands used in our evaluations and the associated generalization gap. As noted in Section 6 of our paper, this is a recognized limitation.
>
> **Introducing a wide variety of dexterous hand models is an important yet separate challenge that has not been addressed** in current research. For comparison, even in the more mature field of cross-embodiment learning for robot arms, state-of-the-art works [1,2,3] are generally limited to about 10 arm types. For dexterous hands, the availability of models is even more constrained. Current methods [4,5,6] focus on morphology-wise variations (e.g., modifying link lengths or structures) to generate robot variants, but designing entirely novel hand models remains a largely unexplored area.
>
> Our work primarily focuses on cross-embodiment training algorithms, but introducing a wide variety of dexterous hands would be a significant future effort that goes beyond the scope of this paper.
>
> To address your concern about generalization potential, we have **included additional results in Appendix C.5 that explore the effect of training with more hand models**. As shown in the table below, increasing the number of training hands consistently improves performance on unseen hands:
>
> | Number of Training Hands | 1 | 2 | 3 | 4 |
> |----------|---------|----------|---------|-----------|
> | Success Rate on Unseen Hands |  0.129 | 0.328 | 0.434 | 0.482 |
>
> These findings suggest that scaling up the number of training hands is a promising direction to reduce the generalization gap. However, as noted, the current lack of a wide range of dexterous hand models limits the feasibility of this approach in the short term.
>
> We believe CrossDex represents an exciting step forward in addressing the underexplored problem of cross-embodiment learning for dexterous hands. Both other reviewers have noted the same limitation but have remained positive about the paper's overall contributions. We kindly ask you to consider the current infeasibility of introducing a broader range of hand models and the potential for scaling up in future work, and we hope you might reconsider the score. If there are any additional concerns, we would be glad to address them.
>
> \
> [1] Kim, et al. "OpenVLA: An Open-Source Vision-Language-Action Model." 2024.
>
> [2] Black, et al. "$\pi_0$: A Vision-Language-Action Flow Model for General Robot Control." 2024.
>
> [3] O'Neill, et al. "Open x-embodiment: Robotic learning datasets and rt-x models." 2023.
>
> [4] Patel, Austin, and Shuran Song. "GET-Zero: Graph embodiment transformer for zero-shot embodiment generalization." 2024.
>
> [5] Hejna, et al. "Task-agnostic morphology evolution." 2021.
>
> [6] Liu, et al. "Revolver: Continuous evolutionary models for robot-to-robot policy transfer." 2022.

---

### Official Review · Reviewer_B2t5 · 2024-11-03

**Soundness:** 3
**Presentation:** 3
**Contribution:** 2
**Rating:** 6
**Confidence:** 4

**Summary:**

This paper presents a method for learning a unified policy for dexterous grasping that can transfer to unseen hand morphologies. It achieves this by first designing an observation space that contains only fingertip positions and palm pose, which can be shared between different hands. Then, the action space is based on the eigengrasp from the MANO hand model. By using the shared observation and action space, the authors can train a reinforcement learning policy on four different hands and transfer the learned policy to new hands.

**Strengths:**

This paper presents a clear story on training policies with multiple embodiments. The paper is well-written and illustrates the motivation from human teleoperation well.

The design of the action space (i.e., a linear combination of eigengrasp coefficients) is interesting.

The idea of using a neural network to approximate the retargeting is also interesting.

The authors also show that the policy can successfully transfer to the real world by training a student policy with vision as the input.

**Weaknesses:**

From my understanding, the observation space contains four fingertip positions. It is not discussed how to deal with three or five fingers, or if the method is specifically designed for anthropomorphic hands. If this is an intrinsic assumption or limitation, it should be discussed in the main paper.

The authors design several ablation experiments to show the importance of the proposed observation and action space. However, there is still one design missing: What if we train a policy that directly predicts the fingertip positions and palm positions while keeping the observation space the same as the proposed method? The fingertip positions could then be converted to joint positions (independently for each finger) using inverse kinematics. This can also generalize to new hands as long as we have the kinematics structure of the new hand.

The retargeting mapping network needs training for each morphology, which means that when evaluating a new, unseen hand morphology, it requires extra data from the target hand. I’m concerned that this is not a fair comparison to the baselines, which does not use these data.

It is also concerning that increasing the number of eigengrasp does not significantly improve performance (Figure 4). This might indicate that the learned grasping actions are quite similar and may be limited for future applications such as functional grasping.

The real experiments are also limited; only three successful trials are demonstrated.

**Questions:**

For policy observation, how should different numbers of fingers be handled? From Figure 2, it seems the authors simply consider four fingers of the hand and ignore the others. If the paper only considers anthropomorphic hands with more than four fingers, this should be clearly stated.

Instead of predicting the eigengrasp and performing retargeting mapping, what is the performance of directly predicting fingertip positions?

When evaluating on unseen hands, does the retargeting mapping network use new hand data for training? If this is the case, I believe there should also be a comparison with baselines using those data.

Can you clarify how “MT-Raw-A” makes predictions on the unseen hands? From my understanding, it outputs joint positions for the unseen hands. But what if the unseen hand has a different number of joints compared to the training hands?

Minor questions and suggestions:
* The YCB dataset contains more than 70 objects. Why were only 45 YCB objects used? What was the criterion for selection?

---

> ### Author Response · Authors · 2024-11-24
> **Thanks for your review! Here, we respond to your comments and address the issues. We hope to hear back from you if you have further questions!**
>
> **Q1.** The observation space contains four fingertip positions. How should different numbers of fingers be handled?
>
> **A1.** Thank you for pointing out the need for clarification. In our experiments, we use the concatenated 3D positions of the palm and the first four fingers: thumb, index, middle, and ring. **For five-fingered hands**, we omit the little finger's position. This simplification is practical as the little finger is typically less critical in grasping tasks. **For three-fingered hands**, the method can be generalized by setting the unused finger positions to zero in the observations. We have included an experiment for a three-fingered D'Claw Gripper in Appendix C.6, which achieves great success rates. In principle, our approach **extends to any hands with up to N fingers**, maintaining compatibility by zero-padding unused positions. We have included this discussion in Section 4.2.
>
> **Q2.** What if we train a policy that directly predicts the fingertip positions and palm positions while keeping the observation space the same as the proposed method?
>
> **A2.** Thank you for your suggestion. We have implemented this baseline, named CrossDex-Pos, using the shared observation space of CrossDex and a 15-dimensional action space representing the five fingertip positions relative to the palm position. These actions are mapped to the joint positions of dexterous hands through optimization-based retargeting. Below is the performance comparison of CrossDex-Pos with CrossDex on five randomly sampled YCB objects:
>
> | Method | Training Hands | Unseen Hands |
> |----------------|--------------------|--------------------|
> | CrossDex-Pos |  0.821+-0.034 | 0.423+-0.056 |
> | CrossDex | 0.892+-0.005 | 0.482+-0.043 |
>
> As shown, CrossDex-Pos underperforms compared to CrossDex on both training and unseen hands. This result highlights that while fingertip position control offers a shared action space, it is less effective in achieving high performance and generalization compared to our eigengrasp-based approach.
> We thank you for this valuable suggestion and will incorporate the full results on all YCB objects into the revised paper to strengthen the evaluation of our method.
>
> **Q3.** Does the retargeting mapping network use new hand data for training when evaluating a new hand?
>
> **A3.** We apologize for the misunderstanding. The retargeting network for each hand is designed to replace the inefficient optimization-based retargeting algorithm and is trained on a **fixed, human hand dataset** with annotations automatically generated by the algorithm. In our experiments, we use the DexPilot retargeting algorithm and the GRAB human hand dataset for this purpose. Importantly, the retargeting algorithm is universal and does not require any additional data collection for unseen hands. This ensures that our comparisons to baselines are fair, as **no additional target hand-specific data is utilized** in our evaluation.
>
>
> **Q4.**  Increasing the number of eigengrasp does not significantly improve performance, indicating that the learned grasping actions are quite similar.
>
> **A4.** We respectfully clarify that CrossDex with a single eigengrasp (CrossDex-1E) achieves a success rate of 78.45%, but increasing the number of eigengrasps leads to a **prominent performance gain** of approximately 10%. This improvement highlights the advantage of utilizing higher-dimensional action spaces to better exploit the dexterity of the hands.
> To draw an analogy, while a parallel gripper (analogous to CrossDex-1E) can successfully grasp most daily objects, higher-dimensional dexterous hands (analogous to CrossDex-k-E, where k>1) demonstrate enhanced capabilities for handling more challenging or intricate grasping tasks. This gain underscores the strength of our method in leveraging the additional control degrees offered by dexterous hands.
>
> **Q5.** Limited real experiments.
>
> **A5.** We have updated the [page](https://sites.google.com/view/crossdex) to include sim-to-real demonstrations with five objects, as well as some failure cases. Our purpose is to demonstrate the feasibility of transferring our simulation and algorithm to the real world. While real-world experiments are limited, these examples highlight the potential applicability of our method.

---

> > ### Author Response · Authors · 2024-11-24
> >
> > **Q6.** How "MT-Raw-A" makes predictions on the unseen hands?
> >
> > **A6.** In addition to arm actions, this baseline predicts the joint positions for each hand, which are padded to 22 dimensions (matching the ShadowHand's DoFs, the highest among all the hands). For an n-DoF hand, the first n dimensions of the output action (ranging from -1 to 1) are scaled to fit the joint limits of the specific hand and sent to the PD controller. Importantly, the URDFs of all dexterous hands are carefully aligned to ensure consistent arrangements of joint position elements (from the thumb to the little finger) and corresponding movements (e.g., opening the hand with increasing values, closing with decreasing values). This alignment enables the baseline to exhibit some degree of generalization to unseen hands.
> >
> > **Q7.** Why were only 45 YCB objects used?
> >
> > **A7.** The selection of 45 YCB objects is related to several reasons: (1) some objects cannot be modeled as rigid bodies, such as chains; (2) some objects contain too many convex hulls after VHACD decomposition, exceeding the capacity of the IsaacGym simulation; (3) some objects are too large or flat, requiring **functional grasping rewards** [1], which are beyond the scope of our study. Despite these exclusions, the remaining 45 objects (see Figure 5) represent a diverse range of daily objects and can be effectively trained with a simple reward function. These details are elaborated in Appendix B.1.
> >
> > [1] Wu, et al. "Unidexfpm: Universal dexterous functional pre-grasp manipulation via diffusion policy." (2024).

---

> > > ### Comment · Reviewer_B2t5 · 2024-11-25
> > >
> > > Thanks for the detailed reply. Most of my concerns have been resolved, and I have increased my score accordingly.
> > >
> > > Regarding Q4, I acknowledge that the performance on training hands improves as the number of eigengrasps increases from 1 to 9. However, my concern is that the performance on unseen hands does not improve, as the results are approximately the same for 1 eigen grasp and 36 eigengrasps.

---

> > > > ### Author Response · Authors · 2024-11-27
> > > >
> > > > Thank you for your review and the positive feedback on our work. Regarding Q4, we acknowledge that while increasing the number of eigengrasps improves performance on training hands by better utilizing the dexterity, the primary bottleneck for zero-shot generalization lies in the generalization gap rather than the degree of dexterity utilization. This limitation explains why increasing the number of eigengrasps does not enhance zero-shot performance. We believe that future work could address this by incorporating a larger training set of diverse dexterous hands to bridge the generalization gap.

---

### Official Review · Reviewer_hjEm · 2024-11-04

**Soundness:** 3
**Presentation:** 4
**Contribution:** 3
**Rating:** 6
**Confidence:** 4

**Summary:**

This paper learns a unified vision-based reinforcement learning policy via teacher-student learning to control various dexterous hands for grasping different objects. The authors propose a unified observation space across various hands, and a universal action space inspired by human hand teleoperation. The learned policy can generalize to unseen dexterous hands in a zero-shot manner.

**Strengths:**

- The authors conducted a comprehensive benchmark and ablation study.
- The presentation of the paper is clear and well-structured.

**Weaknesses:**

- Only LEAP Hand was tested in the real-world experiments.
- Some failure cases should be included in the video.
- Joint position of the arm is included for training policies, which might affect the generalizability to different arms.
- More design choices should be analyzed regarding the impact on generalizability. See the questions below.

**Questions:**

- The authors used four dexterous hands for training and would like to include more different hands in the future. It would be interesting to investigate the impact of the number of hands used for training on the learning performance, e.g., generalizability.
- The dexterous hands for training and testing consist of either 4-fingered or 5-fingered configurations with different DoFs. What would be the impact if for example one or more 3-fingered hand is also included for training?
- The authors randomized the origin position of the joint connecting the hand and the arm to increase robustness and transferability of the policy. What would be the impact of extending this to randomize additional hand parameters?

---

> ### Author Response · Authors · 2024-11-24
> **Thanks for your review! Here, we respond to your comments and address the issues. We hope to hear back from you if you have further questions!**
>
> **Q1.** Only LEAP Hand was tested in the real-world experiments.
>
> **A1.** We apologize for the limited real-world evaluation. We select LEAP Hand because it is affordable and accessible, making it a practical choice for our experiments. We plan to provide sim-to-real results on Inspire Hand in our revision.
>
> **Q2.** Some failure cases should be included in the video.
>
> **A2.** Thank you for your suggestion. We will update the [page](https://sites.google.com/view/crossdex) shortly to include examples of failure cases.
>
> **Q3.** Joint position of the arm is included for training policies, which might affect the generalizability to different arms.
>
> **A3.** We believe that using joint position control for the arm during the RL phase does not limit generalizability to different arms. There are simple solutions converting a joint-based policy to an end-effector policy. For example, during vision-based distillation, we could add a prediction head to predict the next end-effector pose, allowing control of any robotic arm by solving inverse kinematics.
>
> We have also tested learning end-effector control using Euler angles, quaternions, and rotation matrices. However, none of these approaches outperformed direct joint position control, which is why we opted for joint position control in our implementation.
>
> **Q4.** About the impact of the number of hands used for training on the test performance.
>
> **A4.** Thank you for raising this point. We agree that investigating the impact of the number of training hands on generalization is important. To address this, we conducted experiments training CrossDex with different numbers of dexterous hands. The results below demonstrate that increasing the number of training hands consistently improves performance on unseen hands, highlighting the potential for further improvement by including more dexterous hands in the training set:
>
> | Number of Training Hands | 1 | 2 | 3 | 4 |
> |----------|---------|----------|---------|-----------|
> | Success Rate on Unseen Hands |  0.129 | 0.328 | 0.434 | 0.482 |
>
> These results have been added to Appendix C.5.
>
>
> **Q5.** What would be the impact including 3-fingered hands?
>
> **A5.** Thank you for bringing this up. It is indeed valuable to test our method on 3-fingered hands, even though hand retargeting algorithms are less frequently applied to such hands. To explore this, we conducted an experiment using the three-fingered D'Claw Gripper. The retargeting algorithm was adapted by ignoring the ring and small fingers of the human hand and mapping the other three fingers to the robotic fingers. We also used the GRAB dataset to train the retargeting network. We trained the CrossDex policy using the four previous training hands along with the D'Claw Gripper on the five randomly selected YCB objects. The results are presented below:
>
> | Hand |  ShadowHand | Allegro Hand | Ability Hand | Schunk SVH Hand | D'Claw Gripper |
> |----------|---------|----------|---------|-----------|-----|
> | Num. Fingers | 5 | 4 | 5 | 5 | 3 |
> | Success Rate | 0.891 +- 0.009 | 0.892 +- 0.008 | 0.753 +- 0.009 | 0.911 +- 0.009 | 0.853 +- 0.033 |
>
> The results indicate that our method is scalable to the three-fingered D'Claw Gripper, achieving high success rates. These findings, along with additional details, have been added to Appendix C.6.
>
> In Figure 11, we demonstrate that the retargeting network effectively enables the three-fingered hand to open and close, similar to other dexterous hands. However, we observed lower consistency in achieving specific hand poses compared to anthropomorphic hands, due to the significant differences in morphology. For instance, the D'Claw Gripper struggles with certain grasping poses requiring fingers to curve inward. Future work could address these challenges by developing more tailored retargeting methods for hands with distinct morphologies.
>
>
> **Q6.** What would be the impact of randomizing additional hand parameters?
>
> **A6.** We agree that introducing additional randomizations on hand parameters could further enrich the training distribution and improve generalization. Some prior works [1] have explored randomizing link lengths or removing links to create variants of existing hands. However, more extensive randomizations, such as modifying link geometries or overall hand structures, present significant challenges and remain underexplored. We believe future research could focus on developing methods to automatically generate novel dexterous hand models, which would complement and extend the capabilities of CrossDex.
>
> [1] Patel, Austin, and Shuran Song. "GET-Zero: Graph embodiment transformer for zero-shot embodiment generalization." (2024).

---

> > ### Comment · Reviewer_hjEm · 2024-11-26
> >
> > Thank you for providing the additional experiments, which have addressed most of my questions. The proposed approach demonstrates promising results. However, I believe its scalability and generalizability to a wider range of hand morphologies could be explored further. This prevents me from giving a higher score in its current form.

---

> > > ### Author Response · Authors · 2024-11-27
> > >
> > > Thank you for your review and your positive feedback on our work. We fully understand your concerns regarding scalability to a broader range of hands and acknowledge this as a crucial direction for future research. We also recognize this as the primary limitation of the current study and greatly appreciate your acknowledgment of the contributions and potential of our work.

---

### Meta-Review · Area_Chair_f59X · 2024-12-26

**Metareview:**

The authors study the important problem of cross-platform dexterous grasping. Their goal is to learn a grasping policy via RL that works on multiple hands, which they propose to do using a unique universal action space based on the human hand eigengrasps. These can be predicted and translated into robot commands, which means they can be applied to new robots in a zero-shot manner.

Strengths:
The authors do a comprehensive set of ablations to better characterize their method
Reviewers (hjEm, B2t5) agree that it's well written and structured
Interesting method, with a clever universal action space and neural nets for retargeting
Sim-to-real transfer demonstration on a new hand

Weaknesses:
No quantitative sim to real experiments
Limited generalization (potentially due to relatively few hands existing)
Baselines could be stronger
It would be useful to better characterize the variety of grasping strategies being used - success rates do seem somewhat low on unseen hands, and may not improve with more eigengrasps

**Additional Comments On Reviewer Discussion:**

The authors offered detailed rebuttals to the concerns raised by the reviewers, adding some experiments showing how grasp success scaled on new hands as more training hands were added. The authors also added new baselines, such as a fingertip pose based policy. For example, hjEm said their concerns were mostly addressed as a result, as did B2ts.

Reviewer FoGX also raised a number of concerns the authors responded to; but their sticking point was the low success rate on unseen hands, and they did not raise their score. Because of the limited number of hands which exist and the inherent difficulty of the problem, I weighted this review slightly less as a result.

---

### Decision · Program_Chairs · 2025-01-22

Accept (Poster)